# Recent European marine heatwaves are unprecedented but not unexpected

Jamie R. C. Atkins [1,2] ✉, Adam A. Scaife [1,3], Jennifer A. Graham [4], Jonathan Tinker [3] & Paul R. Halloran [1]

The European North-West shelf seas experienced a marine heatwave of unprecedented magnitude in June 2023. Quantifying the likelihood of reoccurrence of similar events is vital for mitigating impacts on marine ecosystems and human activities. Assessing the probability of such events is complicated by climate change-driven changes in the baseline conditions and the short length of the observational record with respect to modes of climate variability. Here, by employing a large ensemble of initialised climate model simulations, we show that the probability of June 2023-like events occurring is approximately 10% in any given year of the present-day climate. Moreover, there has been accelerating growth in the risk of occurrence over the last 30 years. The unprecedented nature of the record-breaking June 2023 event placed European marine heatwaves firmly in the public consciousness. However, the climate change trajectory means that whilst this event was unprecedented, such events should not be unexpected.

Set in the context of a year of exceptional global climate anomalies in 2023[1–3], the European North-West shelf seas (NWS) experienced an episode of notably high ocean temperatures, otherwise known as a marine heatwave (hereafter MHW), in June 2023[4,5] (Fig. 1). The NWS-wide sea surface temperature (SST) anomaly reached +2.9 °C for 16 days relative to the 1982–2012 climatological temperatures for June, which is unprecedented for the satellite era (~1980s onwards)[5]. The Irish Shelf, Celtic Sea and central North Sea sub-regions were hotspots for the event (Fig. 1). Although no impact assessment has yet been published for the June 2023 event itself, MHW activity on the NWS is likely to have biogeochemical consequences and detrimental impacts on marine ecosystems[6,7]. Furthermore, the June 2023 MHW event contributed to record-breaking temperatures and enhanced rainfall over the British Isles[5].

The high-profile June 2023 NWS MHW event has galvanised attention in scientific and public communities and its impacts raise important questions on current levels of risk and preparedness[7,8]. Using the unprecedented June 2023 NWS MHW event as a case study, we ask what the chances are of similar (or stronger) events occurring in summer (June-July-August, hereafter JJA) in the current climate and whether the June 2023 event itself should have been expected in the context of climate change? The urgency of undertaking these tasks is underscored by the already upward-trending frequency and intensity of MHWs on the NWS (with respect to a fixed baseline)[9–12], reflective of global heatwave trends, both atmospheric and marine, detected over recent decades and projected into the future[13–18].

The task of quantifying present-day risk of extreme events is typically hindered by climate change-driven changes in the baseline conditions, the availability of a relatively short satellite era (≈1980s onwards) observational record and the fact that observations represent in effect just a singular realisation of the climate system. Here, instead of relying on the observational record, we pool realisations from an ensemble of initialised climate model simulations (UK Met Office Global Seasonal Forecasting System hindcasts, see Methods) to generate a large distribution of plausible summer NWS MHW events which could occur in the present day. The distribution generated is 60 times the size of the satellite era observational record ($n = 2520$) and can be used to calculate robust statistics on current MHW likelihoods when shown to produce realistic simulations. This approach has been applied previously to atmospheric extreme events[17,19–24] and is commonly referred to as UNSEEN (UNprecedented Simulated Extremes using Ensembles)[21]. To our best knowledge, this method has not previously been applied to marine events.

## Results

### An UNSEEN approach to NWS summer MHW risk

The unprecedented June 2023 MHW event reached Category II intensity (according to the Hobday et al. MHW classification scheme[25]) for approximately two weeks on the NWS[5]. Therefore, to focus the analysis on June 2023-like events, we define the target event for UNSEEN as the peak 14-day rolling mean JJA SST anomaly. Hereafter, we refer to these as JJA-

[1]Faculty of Environment, Science and Economy, University of Exeter, Exeter, UK. [2]Institute for Marine and Atmospheric Research, Utrecht University, Utrecht, The Netherlands. [3]UK Met Office Hadley Centre, Exeter, UK. [4]Centre for Environment, Fisheries and Aquaculture Science (Cefas), Lowestoft, UK. ✉ e-mail: j.r.c.atkins@uu.nl

**Fig. 1 | The June 2023 European North-West shelf seas (NWS) marine heatwave. a** NWS sub-regions, for which SST sub-regional means are calculated for analysis throughout this study. Sub-region names are marked by text annotations. Hatching marks sub-regions in which the UNSEEN protocol is applied here. **b** Observational estimates of June 2023 SST anomalies on the NWS relative to 1993–2016 climatological June. Black line marks the 200 m isobath. **c** timeseries of observational estimate for summer (JJA) NWS-mean SST in 2023 (red line) vs. observational 1993–2016 JJA SST mean and 90th percentile (dashed and dotted black lines, respectively). Dark grey shading highlights June period.

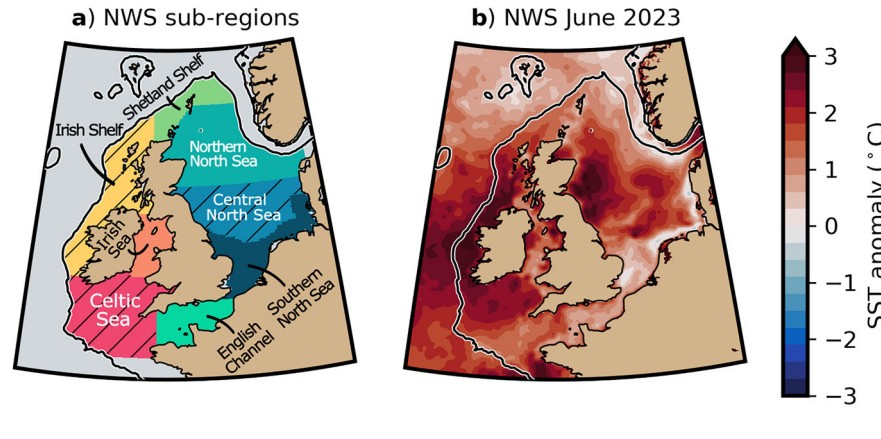

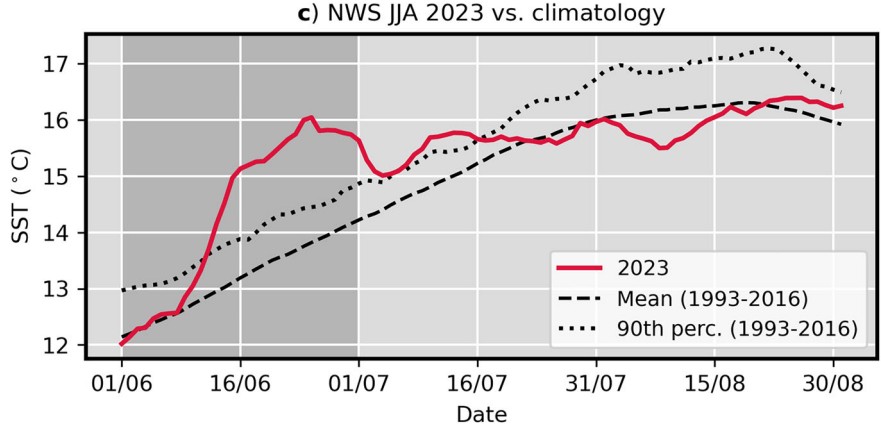

14MAX events. For the model simulations, JJA-14MAX events are calculated for all years and all ensemble members across a hindcast period. This produces a sample of 2520 simulated events (24 years × 105 members). Note, the underlying SST data are detrended by fitting a linear least-squares trend and adjusting each point by the difference between its fitted value and the fitted value at the series end (2024). We refer to this procedure as reference-year detrending (see Methods). This is because there are significant linear historical warming trends in the Celtic Sea, central North Sea and Irish shelf sub-regions (≈0.022 °C, ≈0.040 °C and ≈0.021 °C year$^{-1}$, respectively; Fig. S1) which must be accounted for to make the distributions reflective of present-day climate.

We apply the UNSEEN protocol to the Celtic Sea, central North Sea and Irish Shelf sub-regions. These sub-regions experienced especially pronounced SST anomalies during the June 2023 NWS MHW[5] (Fig. 1b) and their assessment is therefore timely. Moreover, individual ensemble members' JJA simulations in the Celtic Sea, central North Sea and Irish Shelf have the necessary dispersion at 2–4 months lead time for the UNSEEN approach to work[20,26] (see also Methods). To elaborate, the evolution of summer SST in these seasonally stratifying sub-regions is strongly forced by the atmosphere[27]. This means that simulated SST quickly decouples from initial ocean conditions and ensemble members spread due to the influence of chaotic atmospheric variability. Other sub-regions of the NWS (e.g. English Channel and southern North Sea) display a high degree of memory from initial conditions on monthly timescales in summer (i.e. persistence)[27]. The UNSEEN method cannot be applied in these situations as the simulations from individual members are highly dependent (correlated), which reduces the effective sample size of the distribution.

In line with the UNSEEN protocol, it is imperative to assess model fidelity before event statistics can be calculated (see Methods). That is, the ability of the model to adequately simulate realistic alternative realisations of the target event, and that the observed record can reliably be drawn from the distribution of model states. For the Celtic Sea and central North Sea sub-

regions, UNSEEN model fidelity tests show that the mean, standard deviation, skewness and kurtosis of the observational timeseries all lie within the central 95% of the modelled distributions in both sub-regions (Fig. 2). Thus, the model passes the UNSEEN model fidelity tests for JJA-14MAX events in the Celtic Sea and central North Sea, and there is no need for any bias correction. The model distributions are statistically indistinguishable from and interchangeable with observational estimates. In the Irish Shelf sub-region, however, the model fails model fidelity tests (observational standard deviation lies outside the central 95% of the modelled distribution; Fig. 2f). As such, the model distribution is unsuitable for UNSEEN analysis in the Irish Shelf sub-region and subsequent analysis is performed for the Celtic Sea and central North Sea sub-regions only.

### Unprecedented MHWs on the NWS
Real-time observational estimates recorded the NWS-wide peak 14-day rolling mean SST anomaly in June 2023 at +2.13 °C, expressed as de-seasonalised anomalies against 1993–2016 climatology (the model hindcast period). This was unprecedented in the satellite era and substantially higher than the previous record (+1.74 °C in 2009). At the sub-region level, 2023 JJA-14MAX anomalies were recorded at +3.12 °C and +2.56 °C for the central North Sea and Celtic Sea, respectfully. Using the UNSEEN distributions of JJA-14MAX events (Fig. 3a and b), the probability of similar or greater magnitude of SST anomalies (i.e. a June 2023-like event) occurring within any one year in present-day climate (2024) is estimated to be 13.8% in the Celtic Sea (95% confidence interval 12.5% to 16.3%) and 9.8% in the central North Sea (8.6% to 11.0%). The likelihoods decrease as a function of strength of the event anomaly (Fig. 3c and d) but still remain sizeable for events 0.5 °C greater than June 2023 (5.4% probability in the Celtic Sea and 3.9% in the central North Sea) and even 1 °C greater than June 2023 (2.1% in the Celtic Sea and 1.4% in the central North Sea). Our analysis suggests that there is a near-zero probability of events greater than 2 °C above the current record occurring across both sub-regions in present-day climate.

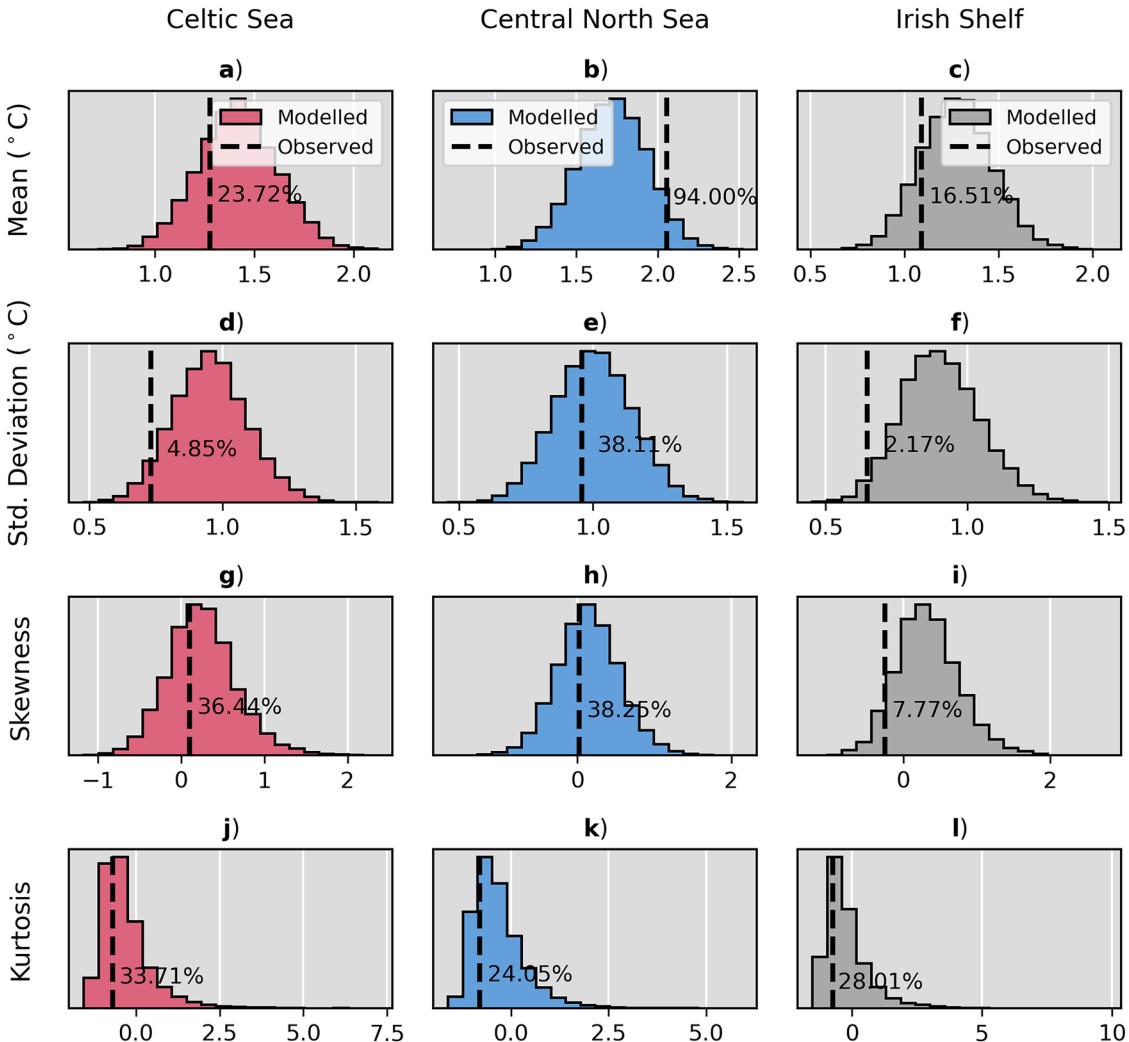

**Fig. 2 | UNSEEN fidelity tests for MHWs.** Frequency distributions of proxy model JJA-14MAX timeseries (*n* = 10,000) mean, standard deviation, skewness and kurtosis (panel rows) in the Celtic Sea (red histograms; panel column **a**, **d**, **g**, **j**), central North Sea (blue histograms; column **b**, **e**, **h**, **k**) and Irish Shelf (grey histograms; column **c**, **f**, **i**, **l**) sub-regions. The observed timeseries statistic for each statistical moment is marked by black dashed lines. The percentile value at which the observed statistic lies within the modelled distribution is marked by text annotation. See Methods for more information on UNSEEN fidelity testing.

By iterating the year of the linear fit about which the underlying SST data are transformed, as part of the UNSEEN reference-year detrending methodology to adjust the modelled distribution to the climate of a given year (see Methods), it is possible to estimate the probability of a June 2023-like event occurring as a function of time over recent decades, or in other words as a function of the background warming trend. There is a clear and rapid nonlinear growth in risk of June 2023-like events occurring in the Celtic Sea and central North Sea sub-regions over recent decades (Fig. 3e and f). In 1993, there was an estimated 3.8% (95% confidence interval 3.0% to 4.6%) and 0.7% (0.4% to 1.0%) chance of a June 2023-like event occurring in each sub-region. In 2023, i.e. the time of the actual event, the estimates suggest the probabilities grew to 13.4% (12.1% to 14.9%) and 8.8% (7.8% to 10.0%), respectively. This implies that the background temperature trend has contributed substantially to MHW risk on the NWS and made the actual June 2023 MHW event ≈ 3.5 times more likely in the Celtic Sea and ≈ 12.5 times more likely in the central North Sea relative to 30 years prior.

## Discussion

UNSEEN estimates of summer MHW risk reveal a strikingly high likelihood of June 2023-like events occurring in present-day climate on the NWS. Indeed, the likelihood has grown rapidly over the last 30 years, reflective of the tendency for heatwave development, both atmospheric and marine, to be amplified by the climate change trend[13–16,28]. Nevertheless, the nonlinear increase in extreme event likelihood in response to a linear warming trend in SST is to be expected given that a positive shift in the mean of a Gaussian distribution results in disproportionately increased probabilities of the upper tail[29,30]. In further support of our findings, recent work[31] applies a different attribution method, namely counterfactual simulations using unconstrained model ensembles, but for the wider North Atlantic basin, and similarly argues that 2023-type events should be expected today (≈10-year return period) in the context of anthropogenic warming. A different study[5] has also demonstrated a contribution of the warming trend to the intensity of SST anomalies during the NWS June 2023 MHW and that such levels are projected to become commonplace past the mid-century under RCP8.5 conditions. On a global scale, additional work[2] has shown using a large sample of observation-based synthetic SST timeseries that record-breaking anomalies in 2023–2024 would have been practically impossible without a global warming trend. Our study complements and builds upon previous attribution studies by using observationally constrained models to provide actionable near-term information which will be critical to immediate on-shelf marine planning and management. Moreover, to our best knowledge, the accelerating rate of change in probability of unprecedented NWS MHWs over recent decades is shown here for the first time.

**Fig. 3 | UNSEEN MHWs exceeding the June 2023 event.** For the Celtic Sea (red; panel column **a**, **c**, **e**) and central North Sea (blue; panel column **b**, **d**, **f**): UNSEEN distribution of modelled JJA-14MAX events (**a**, **b**), UNSEEN estimate of event chance as a function of degrees Celsius above June 2023 (**c**, **d**) and UNSEEN estimate of June 2023 event chance as a function of time since 1993 (**e**, **f**). Observed peak 14-day rolling mean SST in June 2023 is marked by the black dashed lines in (**a**) and (**b**), and the probability of this event in the UNSEEN distribution (i.e. 100% minus percentile rank) is shown by text annotation. In **c**–**f** probabilities are also calculated relative to observational estimates, shading represents the 95% confidence interval calculated by bootstrapping ($n = 1000$). The UNSEEN reference-year detrending approach for 2024 climate (see Methods) is applied in all panels except (**e**) and (**f**) where probabilities are calculated from distributions which are reference-year detrended to reflect the climate of each year in turn. The black dashed lines in (**e**) and (**f**) mark the year 2023, i.e. when the June 2023 event actually occurred.

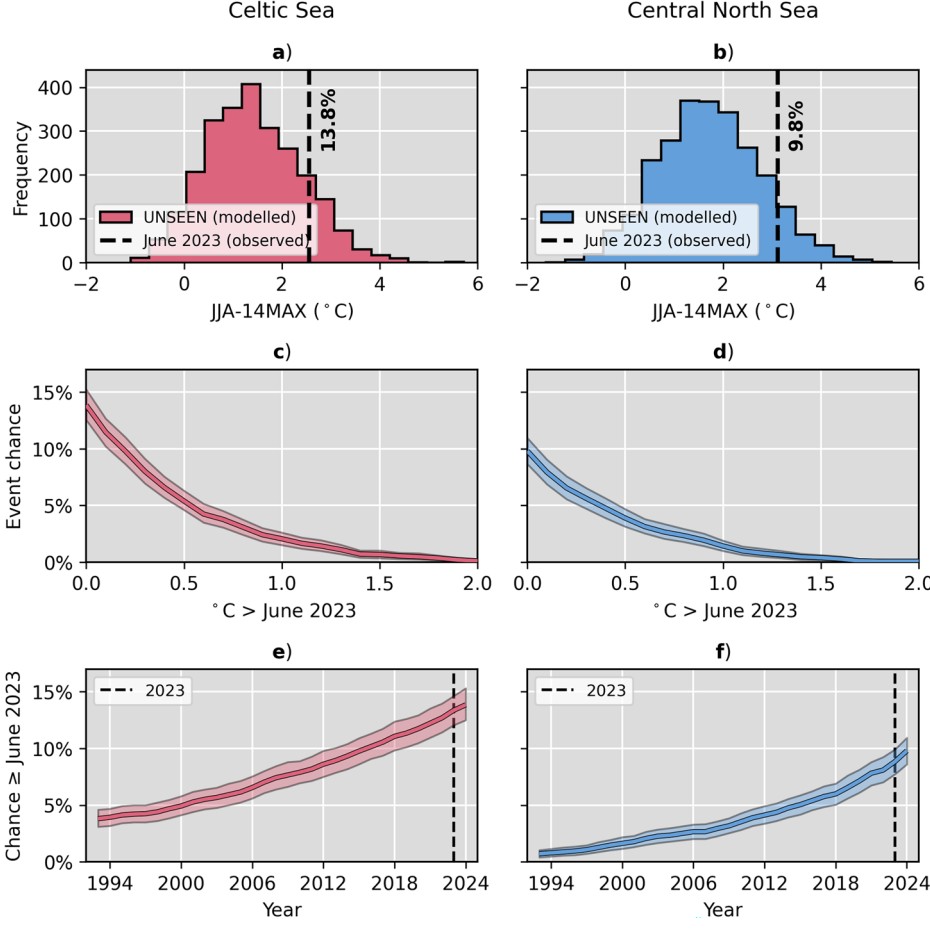

The likelihood of June 2023-like events occurring in the Celtic Sea is estimated to be substantially higher (≈4% higher) than the central North Sea in present-day climate and is consistently higher when estimated for each of the last 30 years. UNSEEN fidelity tests suggest that distributions of the modelled JJA-14MAX timeseries means in the central North Sea (Celtic Sea) are somewhat cold (warm) biased relative to the observed statistic, even if they are shown to be statistically indistinguishable according to the UNSEEN fidelity protocol. In effect, the model may tend towards simulating lower-strength events in the central North Sea than we may expect in reality. This could explain some of the difference between Celtic Sea and central North Sea probabilities extracted from the UNSEEN distribution. However, it is unlikely to be the full explanation. Sub-region dependent dynamical drivers of MHW activity could also contribute to different MHW characteristics. Central North Sea MHW activity is modulated predominantly by atmospheric forcing[32,33] whereas sub-regions in the outer portions of the NWS, including the Celtic Sea, tend to have additional influence from open ocean advection[12,34]. This may point to a role of North Atlantic Ocean variability in further enhancing MHW activity in the Celtic Sea.

Marine ecosystems and fisheries globally have been shown to be vulnerable to extreme heat events[35–40]. Previous studies have linked MHW events around the United Kingdom to harmful algal blooms, fish and shellfish landing rates and increasing risk of infectious diseases harmful to humans[6,7,41,42]. Our findings demonstrate that we are increasingly likely to experience unprecedented events today and not just in future decades as NWS SSTs continue to warm[5,43]. Our study demonstrates the immediate need for focused impact studies on NWS ecosystem responses to MHWs, building on previous calls[5,8]. We also encourage industries and policymakers to factor our UNSEEN probability calculations into operational risk assessments, decision-making and public communications to build resilience and mitigate impacts of strong MHWs on the NWS.

In addition to the biogeochemical impacts, strong MHW events will feedback on weather systems felt by humans over land. For example, the United Kingdom broke its June land temperature record by 0.9 °C in 2023, 0.6 °C of which was found to be attributable to feedbacks from excess heat on the NWS during the June 2023 NWS MHW[5]. Similar effects have been felt in other regions, for example in the contribution of an unprecedented MHW in coastal waters to an atmospheric heatwave over Northern Japan in summer 2023[44]. Our results show substantial present-day risk of events stronger than the June 2023 NWS MHW and highlight an opportunity for oceanic and atmospheric extremes research communities to collaborate to improve understanding of compounding events in a changing climate.

In conclusion, our UNSEEN estimates show that there is a high likelihood of June 2023-like events occurring in the Celtic Sea and central North Sea in present-day climate (≈10% or greater chance each summer dependent on sub-region). The chance of even stronger events is nontrivial, and the risk of similar events has grown rapidly over recent decades. Indeed, we estimate that by the time of the actual June 2023 event the likelihood of occurrence had near quadrupled since 1993 in the Celtic Sea, whilst the probability increased by over 12 times in the central North Sea, attributable to the warming trend over the last 30 years. Therefore, the June 2023 NWS MHW event was unprecedented but should not have been unexpected, and more frequent and more intense events can be expected in the near-future in response to committed SST rises.

## Methods
### Model and observations
We use large ensemble initialised climate model realisations of daily NWS SST spanning the period 1993–2016 from operational hindcasts using versions 5 and 6 of the UK Met Office Global Seasonal Forecasting System (GloSea)[45]. Both versions use the HadGEM3 ocean-atmosphere-land-sea-ice coupled climate model. GloSea6 uses HadGEM3-GC3[46] whereas

GloSea5 uses HadGEM3-GC2[47], but the two versions show similar performance. Daily SST data are available only on the native atmospheric grid and not from the GloSea ocean component. Therefore, the spatial resolution of the SST data is at N216 resolution (0.83° in latitude and 0.5° in longitude). The model is initialised with observational analyses of the atmosphere, land, ocean and sea ice and integrated forward for six simulated months. Seven individual members (differing by stochastic perturbation of atmospheric initial conditions[48]) are initialised across three start dates centred on the 1st of the month preceding the summer (JJA) season (25th April, 1st May, 9th May; i.e. lagged mode) and are combined to obtain an ensemble of simulations at 2–4 months lead time for each hindcast year. Therefore, the number of members per hindcast year is 21 (7 members × 3 start dates). Multiple hindcasts have been run over recent years and five are combined here to form a larger ensemble of 105 members (5 × 21-member ensembles) per year. In total, with 24 years of hindcast data, there are 2,520 summer realisations (24 × 105) from which to calculate MHW statistics, which is 60 times the collection of satellite era observed summers (42).

GloSea hindcast data have been used in previous UNSEEN studies in the context of atmospheric extremes[49,50]. Using a multi-model ensemble for UNSEEN analysis[51] would reduce sensitivity to model structure[26] and represents a worthwhile step for future work, but is beyond the scope of the present study.

We use the Operational SST and Sea Ice Analysis system[52,53] (hereafter OSTIA) to provide observational-based estimates of the case study June 2023 MHW event and for UNSEEN model fidelity tests (more detail in the subsequent Methods sub-section). To span the full 1982–2024 observational period (i.e. satellite era), we concatenate two versions of the OSTIA record. That is, the combination of OSTIA near real-time SST fields (2022–near-present) on a 0.05° × 0.05° grid generated from in-situ and satellite observations (infrared and microwave radiometers) plus an OSTIA climate version (1982–2022) on the same grid but produced using coarser temporal resolution observations[53].

All SST data in this study are presented as daily NWS sub-regional means, for locations shown in Fig. 1a. The sub-regions span the areas where the June 2023 NWS MHW was strongest, namely the Irish Shelf, Celtic Sea and central North Sea regions.

### Ensemble member independence and UNSEEN model fidelity tests

GloSea hindcasts are well suited for UNSEEN application. Once initialised, the individual ensemble members are integrated forward beyond the deterministic range of near-term NWS SST forecasting in strongly atmospherically forced NWS sub-regions such as the Celtic Sea and central North Sea in summer[27]. This creates the necessary spread of ensemble members to generate independent/dispersive but equally plausible SST realisations, provided model fidelity tests are passed (see below). Without sufficient dispersion the members are dependent (correlated), and the effective sample size of the UNSEEN distribution is then reduced.

The UNSEEN model fidelity procedure tests that realisations in the modelled distribution (GloSea) and observations (OSTIA) are statistically indistinguishable. To do so, different moments of modelled and observed distributions, i.e. the mean, standard deviation, kurtosis and skewness, are compared[21,22]. We sample the full GloSea ensemble across the hindcast period by bootstrapping (with replacement) to produce 10,000 proxy timeseries of the same length as the observed record over 1993–2016 (common period between model and observations). When the single observed (OSTIA) statistic for each statistical moment lies within the central 95% of the modelled distribution (i.e. within the 2.5th and 97.5th percentiles), we consider the model to be statistically indistinguishable from the observations and therefore suitable for UNSEEN analysis. This approach mirrors that taken in previous studies which apply the UNSEEN approach to climate extremes[17,20–24].

### Reference-year detrending

UNSEEN is intended to assess probabilities of events occurring in present-day climate. Therefore, the impact of a historical warming trend on the underlying SST distribution must be considered. To adjust the GloSea distribution to be centred around a point that represents the present-day climate, we reference-year detrend the full ensemble GloSea data (and OSTIA for fidelity testing) about the end point of the trend, using a similar approach to a previous UNSEEN study on atmospheric heatwaves[17]. The procedure is as follows:

First, for each individual sub-region, we apply a linear regression fit across the 1982–2024 OSTIA JJA SST record (black lines in Fig. S1). Next, the difference between each point on the trend line across 1993–2016 (matching the GloSea hindcast period) and the final point of the linear fit (2024, present day at time of writing; red markers in Fig. S1) is calculated ('last differencing'). We use the 2024 point on the linear fit, as opposed to the actual 2024 data point, to produce a less noisy estimate for 2024. Finally, we subtract the last differenced timeseries from the GloSea hindcast timeseries, thereby transforming the distribution about the 2024 fit. The effect of this reference-year detrending strategy is that the underlying SST data is adjusted from a timeseries of historical SST evolution containing a warming trend to a distribution of alternative SST realisations in the 2024 state of the climate. The linear trend is removed in the reference-year detrended version but, importantly, the internal variability across the hindcast period is preserved. For the purposes of our study, the variability in the hindcast period (1993–2016) is assumed to be an adequate analogue for the present day climate, due to (i) the close temporal proximity between the periods and lack of significant differences in variability and because (ii) the contribution of mean warming to global MHW trends is found to outweigh that of any changes to variability[28,54]. From the new reference-year detrended distribution, we extract the samples of 2024-state events.

The choice of dataset from which the linear trend for the reference-year detrending is calculated is subject to debate. Here, we use the OSTIA trend across 1982–2024 as it is the longest high-quality SST record available. It could be argued that the trend from the GloSea ensemble mean is more suitable, despite it covering a shorter historical period (24 years versus 42 for OSTIA) and requiring extrapolation to 2024, because it samples a greater number of plausible trajectories and suppresses internal variability noise across the timeseries that is inevitably present in the observed trend. In any case, the OSTIA and GloSea-derived trends are statistically indistinguishable (observed trend lies within the central 95% of the model trend distribution; Fig. S2) and result in discrepancies of just ±1–2 percentage points in the final June 2023-like probability estimates for the present day when used for the reference-year detrending (Fig. 3 versus Fig. S3).

## Data availability

Daily GloSea5 and GloSea6 SST hindcast data used in this study (on native grids) were accessed directly from the UK Met Office but are also freely available (interpolated to 1° × 1° grid) from the Copernicus Climate Change Service[55] (https://doi.org/10.24381/cds.181d637e). OSTIA near real-time[56] (https://doi.org/10.48670/moi-00165) and climate[57] (https://doi.org/10.48670/moi-00168) SST datasets are both freely available from the Copernicus Marine Environment Monitoring Service. Post-processed versions of the data are archived at https://doi.org/10.5281/zenodo.17076446.

## Code availability

The code[58] for performing the UNSEEN analysis and producing all figures using post-processed data is available at https://github.com/j-atkins/UNSEEN_MHWs and archived at https://doi.org/10.5281/zenodo.17077016.

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

## Acknowledgements

This work was supported by a GW4 + Doctoral Training Partnership studentship from the Natural Environmental Research Council (NE/S007504/1) and the UK Met Office. AAS and JT were supported by the Met Office Hadley Centre Climate Programme (HCCP) funded by the UK Department for Science, Innovation and Technology (DSIT), the UK Public Weather Service and the UK Department for Environment, Food and Rural Affairs (DEFRA). JAG was supported by the Cefas Seedcorn studentship programme. For the purpose of open access, the author has applied a 'Creative Commons Attribution (CC BY)' licence to any Author Accepted Manuscript version arising from this submission.

## Author contributions

Jamie R. C. Atkins: developed the study design, sourced and collected data, analysed the data and created the figures, led the writing of the manuscript, discussed results, contributed to the manuscript prior to submission. Adam A. Scaife: supervisor, developed the study design, discussed results, contributed to the manuscript prior to submission. Jennifer A. Graham: supervisor, developed the study design, discussed results, contributed to the manuscript prior to submission. Jonathan Tinker: supervisor, sourced and collected data, discussed results, contributed to the manuscript prior to submission. Paul R. Halloran: supervisor, developed the study design, discussed results, contributed to the manuscript prior to submission.

## Competing interests

The authors declare no competing interests.
