## [Transparent Peer Review file · Communications Earth & Environment]

Recent European marine heatwaves are unprecedented but not unexpected

Corresponding Author: Dr Jamie Atkins

Version 0:

Reviewer comments:

Reviewer #1

(Remarks to the Author)

This paper uses a large ensemble of initialized climate model simulations to assess the likelihood of occurrence of extreme warm ocean conditions like the one experienced by the European Northwest Shelf in June 2023. These unprecedented conditions have received considerable attention in the literature, and several efforts have been undertaken to attribute their causes to either climate change or internal variability. In this study, the authors address this issue using a novel approach, the UNSEEN (UNprecedented Simulated Extremes using Ensembles) protocol. This approach is based on initialized climate model simulations, whose large number of realizations allows to overcome the relatively small sample size available from observations. Also, the definition of MHWs in the presence of a trend is a major point of discussion in the MHW community. This paper contributes to that discussion by using a method which is designed to remove the long-term trend and assess probabilities for the present-day period. For these reasons, the paper provides a valuable contribution to this scientific problem.

That said, there are a few major points that the authors should clarify:

1. The trend that is removed with their pivot detrending is linear, while there are indications that the real-world trend may be nonlinear at least in some parts of the world (e.g., Xu et al. 2022). The authors should discuss this issue and the possible implications of the linearity assumption.
2. While the UNSEEN approach is used to assess present-day probabilities, this approach is based on simulations over the period 1993-2016. How can we be sure that the statistics of that period are adequate for present-day conditions?
3. The UNSEEN approach allows an evaluation of the probabilities for present-day conditions, effectively detrending the data. However, the authors also show that the likelihood of these extreme conditions has increased in recent years. If the effect of the mean state is removed from the present-day estimate, what causes the change in probabilities? Has internal variability changed? How? It would be important to include a discussion of this point.

Itemized comments (mainly minor):

Caption of Fig. 1 and line 126. Here there is a mention to the 1993-2016 climatology. Before reading the Methods section it is not obvious why this period is chosen to define the climatology. It is only in the Methods section that we find out that this is the period covered by the initialized climate simulation, so I suggest that the use of this period is clarified either in the caption of Fig. 1 or in line 126 to avoid confusion.

Line 57-58 "upward trending frequency and intensity...". The upward trending frequency and intensity may depend on the definition of MHWs. If the definition is based on a fixed baseline the presence of a warming trend will lead to a more frequent and larger excursion of the SST anomalies above the fixed threshold as discussed in recent papers (e.g., Capotondi et al. 2024; Smith et al. 2025). Since the baseline choice seems central to the UNSEEN methodology, I suggest that the authors clarify this point here, and briefly discuss the issue of defining MHWs in a changing climate.

Line 95. Should “stratifying” be “stratified”?

Line 166. Another useful reference here is Xu et al. (2022). This paper shows the key role played by the warming trend in the increasing frequency of MHWs. It also shows the influence of the nonlinearity of the trend on the assessment of changes in variability.

Line 168. The SST trend is not necessarily linear. Please discuss the implications of this assumption.

Lines 187-189. This sentence is not clear to me. Maybe the content could be expanded and explained in different words or using an example.

References

Capotondi, A. et al. (2024), A Global overview of marine heatwaves in a changing climate, *Communications Earth & Environment*, <https://www.nature.com/articles/s43247-024-01806-9>

Smith, K.E. et al. 2025, Baseline matters: Challenges and implications of different marine heatwave baselines, *Progress Oceanography*, <https://www.sciencedirect.com/science/article/pii/S0079661124002106>

Xu, T., et al. 2022, An increase in marine heatwaves without significant changes in surface ocean temperature variability, *Nat. Comm.*, <https://www.nature.com/articles/s41467-022-34934-x>

Reviewer #2

(Remarks to the Author)

Please see the attachment for my comments.

Reviewer #3

(Remarks to the Author)

This work explores how the likelihood of marine heatwaves (MHWs) in the European North-Western Shelf (NWS) has changed with time and quantifies the probability of a particular record-breaking event occurring. The foundation of this work is the “UNSEEN” method, which builds a distribution of potential MHW events from physically plausible climate model simulations. The UNSEEN methodology is validated, here and elsewhere. The main results show how an event of the magnitude of June 2023 has become more likely, and continues to become more likely, with time over the past 3 decades.

The research is engaging and the content well written. I enjoyed the succinct nature of the paper, although I find that also means some explanations and justifications are lacking. In particular, the UNSEEN methodology needs to be explained carefully. The main conclusion, on the increased likelihood of intense MHWs, is somewhat obvious, but I appreciate that the quantification is an important step. The authors communicate fairly well the need for such studies, but I feel not yet fully convinced that the quantification alone justifies publication in this journal. Before publication some clarifications need to be made, in my opinion.

Main Comments

In the abstract, some statements are unclear. “The probability of June 2023-like events occurring in the present day is approximately 10% in any given year”; given the definition of a MHW as an exceedance of the 90th percentile, in a detrended dataset, the probability of an event is always 10%. Also, what do you mean by “any given year”? I thought the chance increased with time (thanks to the trend).

If this was an unprecedented, what was the next closest event i.e. what is the next largest JJA-14MAX on record?

Recent and similar high-profile work has been published on “unlikely but not unexpected” SST jumps. How does this compare to UNSEEN? Terhaar, J., Burger, F.A., Vogt, L. et al. Record sea surface temperature jump in 2023–2024 unlikely but not unexpected. *Nature* 639, 942–946 (2025). <https://doi.org/10.1038/s41586-025-08674-z>

Trends are an important part of the discussion – please consider adding maps in the supplementary.

The authors claim to use a dataset of “plausible...events which could occur in the present day”. The benefit of this dataset is that it uses a very large ensemble. However, it is based on one model. That the representation of events changes between models should be considered. The authors should explain why using a larger ensemble from one model is better than using a potentially equally-large ensemble from a range of models (i.e. the other Copernicus seasonal forecasting systems).

Where ensemble members agree (in regions of high memory/persistence) the UNSEEN model is not used. The authors state that the effective sample size decreases. Is it not that the distribution simply narrows? Surely the number of samples is the same?

Regarding the fidelity tests, falling with the central 90% of the distribution seems a very easy target. I notice that if the

threshold were 90%, then the Celtic Sea would be removed too. Please justify this better.

Authors effectively communicate the need for risk assessments and such quantifications. How does the UNSEEN method compare to attribution studies which also aim to provide how likelihoods of events change with climate change?

This study focuses on a single NWS MHW, so the inclusion of "Recent... heatwaves" in the title is inaccurate.

Minor Comments

Line 153: Why is the growth non-linear? Is the background SST trend non-linear?

Line 230: Is GloSea run in burst or lagged mode? Or is this a non-operational version?

Line 231: Typically new releases have a range of differences. Is the only difference between 6 and 5 the model? No change in resolution, initial conditions?

Line 244: What are these multiple hindcasts? How do they differ?

Line 246: Where does the factor of two (from the two versions, 5 & 6) come into the calculation of 2520?

Line 290: The last point being 2016?

Line 307: I do not think "not shown" is allowed in this journal. Anyway, I recommended including trend maps.

Decision Letter:

Dear Mr Atkins,

Your manuscript titled "Recent European marine heatwaves are unprecedented but not unexpected." has now been seen by 3 reviewers, and we include their comments at the end of this message. They find your work of interest, but some important points are raised. We are interested in the possibility of publishing your study in Communications Earth & Environment, but would like to consider your responses to these concerns and assess a revised manuscript before we make a final decision on publication.

We therefore invite you to revise and resubmit your manuscript, along with a point-by-point response that takes into account the points raised. Please highlight all changes in the manuscript text file.

Please submit your point-by-point responses as a separate file, distinct from your cover letter where you can add responses to the Editors' comments that you do not want to be made available to the reviewers. Word files are preferred. We recommend that any figures, tables or graphs that are included in the response to reviewers are also included in the main article or Supplementary Information.

Please use the following link to submit your revised manuscript, point-by-point response to the referees' comments (which should be in a separate document to any cover letter), a tracked-changes version of the manuscript (as a PDF file) and the completed checklist:

Link Redacted

We hope to receive your revised paper within six weeks; please let us know if you aren't able to submit it within this time so that we can discuss how best to proceed. If we don't hear from you, and the revision process takes significantly longer, we may close your file. In this event, we will still be happy to reconsider your paper at a later date, as long as nothing similar has been accepted for publication at Communications Earth & Environment or published elsewhere in the meantime.

Please do not hesitate to contact us if you have any questions or would like to discuss these revisions further. We look forward to seeing the revised manuscript and thank you for the opportunity to review your work.

Best regards,

Alienor Lavergne, PhD
Senior Editor, Communications Earth & Environment
Consulting Editor, Communications Sustainability
ORCID: 0000-0002-4591-1217

Springer Nature
The Campus, 4 Crinan Street, London N1 9XW, UK
www.nature.com/commsenv
@commsearth.bsky.social

EDITORIAL POLICIES AND FORMATTING

- Behavioural and social science
- Ecological, evolutionary & environmental sciences
- Life sciences

Furthermore, please align your manuscript with our format requirements, which are summarized on the following checklist: [Communications Earth & Environment formatting checklist](https://www.nature.com/documents/commsj-phys-style-formatting-checklist-article.pdf)

and also in our style and formatting guide [Communications Earth & Environment formatting guide](https://www.nature.com/documents/commsj-phys-style-formatting-guide-accept.pdf) .

***** DATA:** Communications Earth & Environment endorses the principles of the Enabling FAIR data project (<http://www.copdess.org/enabling-fair-data-project/>). We ask authors to make the data that support their conclusions available in permanent, publically accessible data repositories. (Please contact the editor if you are unable to make your data available).

All Communications Earth & Environment manuscripts must include a section titled "Data Availability" at the end of the Methods section or main text (if no Methods). More information on this policy, is available at <http://www.nature.com/authors/policies/data/data-availability-statements-data-citations.pdf>.

If a community resource is unavailable, data can be submitted to generalist repositories such as [figshare](https://figshare.com/) or [Dryad Digital Repository](http://datadryad.org/). Please provide a unique identifier for the data (for example a DOI or a permanent URL) in the data availability statement, if possible. If the repository does not provide identifiers, we encourage authors to supply the search terms that will return the data. For data that have been obtained from publically available sources, please provide a URL and the specific data product name in the data availability statement. Data with a DOI should be further cited in the methods reference section.

REVIEWER COMMENTS:

Reviewer #1 (Remarks to the Author):

This paper uses a large ensemble of initialized climate model simulations to assess the likelihood of occurrence of extreme warm ocean conditions like the one experienced by the European Northwest Shelf in June 2023. These unprecedented conditions have received considerable attention in the literature, and several efforts have been undertaken to attribute their causes to either climate change or internal variability. In this study, the authors address this issue using a novel approach, the UNSEEN (UNprecedented Simulated Extremes using Ensembles) protocol. This approach is based on initialized climate model simulations, whose large number of realizations allows to overcome the relatively small sample size available from observations. Also, the definition of MHWs in the presence of a trend is a major point of discussion in the MHW community. This paper contributes to that discussion by using a method which is designed to remove the long-term trend and assess probabilities for the present-day period. For these reasons, the paper provides a valuable contribution to this scientific problem.

That said, there are a few major points that the authors should clarify:

1. The trend that is removed with their pivot detrending is linear, while there are indications that the real-world trend may be nonlinear at least in some parts of the world (e.g., Xu et al. 2022). The authors should discuss this issue and the possible implications of the linearity assumption.
2. While the UNSEEN approach is used to assess present-day probabilities, this approach is based on simulations over the period 1993-2016. How can we be sure that the statistics of that period are adequate for present-day conditions?
3. The UNSEEN approach allows an evaluation of the probabilities for present-day conditions, effectively detrending the data. However, the authors also show that the likelihood of these extreme conditions has increased in recent years. If the effect of the mean state is removed from the present-day estimate, what causes the change in probabilities? Has internal variability changed? How? It would be important to include a discussion of this point.

Itemized comments (mainly minor):

Caption of Fig. 1 and line 126. Here there is a mention to the 1993-2016 climatology. Before reading the Methods section it is not obvious why this period is chosen to define the climatology. It is only in the Methods section that we find out that this is the period covered by the initialized climate simulation, so I suggest that the use of this period is clarified either in the caption of Fig. 1 or in line 126 to avoid confusion.

Line 57-58 “upward trending frequency and intensity...”. The upward trending frequency and intensity may depend on the definition of MHWs. If the definition is based on a fixed baseline the presence of a warming trend will lead to a more frequent and larger excursion of the SST anomalies above the fixed threshold as discussed in recent papers (e.g., Capotondi et al. 2024; Smith et al. 2025). Since the baseline choice seems central to the UNSEEN methodology, I suggest that the authors clarify this point here, and briefly discuss the issue of defining MHWs in a changing climate.

Line 95. Should “stratifying” be “stratified”?

Line 166. Another useful reference here is Xu et al. (2022). This paper shows the key role played by the warming trend in the increasing frequency of MHWs. It also shows the influence of the nonlinearity of the trend on the assessment of changes in variability.

Line 168. The SST trend is not necessarily linear. Please discuss the implications of this assumption.

Lines 187-189. This sentence is not clear to me. Maybe the content could be expanded and explained in different words or using an example.

References

Capotondi, A. et al. (2024), A Global overview of marine heatwaves in a changing climate, *Communications Earth & Environment*, <https://www.nature.com/articles/s43247-024-01806-9>

Smith, K.E. et al. 2025, Baseline matters: Challenges and implications of different marine heatwave baselines, *Progress Oceanography*, <https://www.sciencedirect.com/science/article/pii/S0079661124002106>

Xu, T., et al. 2022, An increase in marine heatwaves without significant changes in surface ocean temperature variability, *Nat. Comm.*, <https://www.nature.com/articles/s41467-022-34934-x>

Reviewer #2 (Remarks to the Author):

Please see the attachment for my comments.

Reviewer #3 (Remarks to the Author):

This work explores how the likelihood of marine heatwaves (MHWs) in the European North-Western Shelf (NWS) has changed with time and quantifies the probability of a particular record-breaking event occurring. The foundation of this work is the “UNSEEN” method, which builds a distribution of potential MHW events from physically plausible climate model simulations. The UNSEEN methodology is validated, here and elsewhere. The main results show how an event of the magnitude of June 2023 has become more likely, and continues to become more likely, with time over the past 3 decades.

The research is engaging and the content well written. I enjoyed the succinct nature of the paper, although I find that also means some explanations and justifications are lacking. In particular, the UNSEEN methodology needs to be explained carefully. The main conclusion, on the increased likelihood of intense MHWs, is somewhat obvious, but I appreciate that the quantification is an important step. The authors communicate fairly well the need for such studies, but I feel not yet fully convinced that the quantification alone justifies publication in this journal. Before publication some clarifications need to be made, in my opinion.

Main Comments

In the abstract, some statements are unclear. "The probability of June 2023-like events occurring in the present day is approximately 10% in any given year"; given the definition of a MHW as an exceedance of the 90th percentile, in a detrended dataset, the probability of an event is always 10%. Also, what do you mean by "any given year"? I thought the chance increased with time (thanks to the trend).

If this was an unprecedented, what was the next closest event i.e. what is the next largest JJA-14MAX on record?

Recent and similar high-profile work has been published on "unlikely but not unexpected" SST jumps. How does this compare to UNSEEN? Terhaar, J., Burger, F.A., Vogt, L. et al. Record sea surface temperature jump in 2023–2024 unlikely but not unexpected. *Nature* 639, 942–946 (2025). <https://doi.org/10.1038/s41586-025-08674-z>

Trends are an important part of the discussion – please consider adding maps in the supplementary.

The authors claim to use a dataset of "plausible...events which could occur in the present day". The benefit of this dataset is that it uses a very large ensemble. However, it is based on one model. That the representation of events changes between models should be considered. The authors should explain why using a larger ensemble from one model is better than using a potentially equally-large ensemble from a range of models (i.e. the other Copernicus seasonal forecasting systems).

Where ensemble members agree (in regions of high memory/persistence) the UNSEEN model is not used. The authors state that the effective sample size decreases. Is it not that the distribution simply narrows? Surely the number of samples is the same?

Regarding the fidelity tests, falling with the central 90% of the distribution seems a very easy target. I notice that if the threshold were 90%, then the Celtic Sea would be removed too. Please justify this better.

Authors effectively communicate the need for risk assessments and such quantifications. How does the UNSEEN method compare to attribution studies which also aim to provide how likelihoods of events change with climate change?

This study focuses on a single NWS MHW, so the inclusion of "Recent... heatwaves" in the title is inaccurate.

Minor Comments

Line 153: Why is the growth non-linear? Is the background SST trend non-linear?

Line 230: Is GloSea run in burst or lagged mode? Or is this a non-operational version?

Line 231: Typically new releases have a range of differences. Is the only difference between 6 and 5 the model? No change in resolution, initial conditions?

Line 244: 7What are these multiple hindcasts? How do they differ?

Line 246: Where does the factor of two (from the two versions, 5 & 6) come into the calculation of 2520?

Line 290: The last point being 2016?

Line 307: I do not think "not shown" is allowed in this journal. Anyway, I recommended including trend maps.

** Visit Nature Portfolio's author and referees' website at <http://www.nature.com/authors> for information about policies, services and author benefits**

Communications Earth & Environment is committed to improving transparency in authorship. As part of our efforts in this direction, we are now requesting that all authors identified as 'corresponding author' create and link their Open Researcher and Contributor Identifier (ORCID) with their account on the Manuscript Tracking System prior to acceptance. ORCID helps the scientific community achieve unambiguous attribution of all scholarly contributions. You can create and link your ORCID from the home page of the Manuscript Tracking System by clicking on 'Modify my Springer Nature account' and following the instructions in the link below. Please also inform all co-authors that they can add their ORCID to their accounts and that they must do so prior to acceptance.

Version 1:

Reviewer comments:

Reviewer #1

(Remarks to the Author)

The authors have satisfactorily addressed all of my comments, and I think that the clarity of the paper has improved. I'm happy to recommend acceptance.

Reviewer #2

(Remarks to the Author)

All my comments have been addressed.

Reviewer #3

(Remarks to the Author)

The authors have responded to all my comments and queries comprehensively. The paper I believe is ready and deserving of publication in this journal. I have no further comments or corrections. Congratulations on this work.

Decision Letter:

Dear Dr Atkins,

Your manuscript titled "Recent European marine heatwaves are unprecedented but not unexpected" has now been seen by our reviewers, whose comments appear below. In light of their advice we are delighted to say that we are happy, in principle, to publish a suitably revised version in Communications Earth & Environment.

We ask that you edit your manuscript to comply with our format requirements and to maximise the accessibility and therefore the impact of your work.

EDITORIAL REQUESTS:

****Please take care to match our formatting and policy requirements. We will check revised manuscript and return manuscripts that do not comply. Such requests will lead to delays. ****

SUBMISSION INFORMATION:

OPEN ACCESS:

Communications Earth & Environment is a fully open access journal. Articles are made freely accessible on publication. For further information about article processing charges, open access funding, and advice and support from Nature Portfolio, please visit <https://www.nature.com/commsenv/open-access>

Link Redacted

Best regards,
Nicola

Nicola Colombo
Associate Editor, Communications Earth & Environment
Consulting Editor, Communications Sustainability

REVIEWERS' COMMENTS:

Reviewer #1 (Remarks to the Author):

The authors have satisfactorily addressed all of my comments, and I think that the clarity of the paper has improved. I'm happy to recommend acceptance.

Reviewer #2 (Remarks to the Author):

All my comments have been addressed.

Reviewer #3 (Remarks to the Author):

The authors have responded to all my comments and queries comprehensively. The paper I believe is ready and deserving of publication in this journal. I have no further comments or corrections. Congratulations on this work.

** Visit Nature Portfolio's author and referees' website at www.nature.com/authors for information about policies, services and author benefits**

Response to reviewers

We are very grateful for the careful assessment and insightful comments from the three reviewers. All reviewers have raised constructive suggestions which we have addressed comprehensively. Across the remainder of this document, we have copied the full text from each reviewer (in black text) and respond to each point individually (blue text).

Please note that we have also made some small edits to the text throughout the manuscript, in addition to the requests of the reviewers, to further enhance readability and clarity. We would like to draw attention to one change in particular. That is, we now refer to the detrending procedure employed in our study as "reference-year detrending" throughout the manuscript (previously "pivot detrending"). Multiple reviewers have raised aspects of the detrending procedure, and we now use this more descriptive term to enhance clarity (in addition to our responses to the reviewers' specific comments).

Where we refer to line numbers in this reviewer response document, we refer to the line numbers as they appear in the new revised manuscript document, unless otherwise stated.

Reviewer #1

This paper uses a large ensemble of initialized climate model simulations to assess the likelihood of occurrence of extreme warm ocean conditions like the one experienced by the European Northwest Shelf in June 2023. These unprecedented conditions have received considerable attention in the literature, and several efforts have been undertaken to attribute their causes to either climate change or internal variability. In this study, the authors address this issue using a novel approach, the UNSEEN (UNprecedented Simulated Extremes using Ensembles) protocol. This approach is based on initialized climate model simulations, whose large number of realizations allows to overcome the relatively small sample size available from observations. Also, the definition of MHWs in the presence of a trend is a major point of discussion in the MHW community. This paper contributes to that discussion by using a method which is designed to remove the long-term trend and assess probabilities for the present-day period. For these reasons, the paper provides a valuable contribution to this scientific problem.

Thank you for the encouraging comments and useful suggestions. A point-by-point response follows below:

That said, there are a few major points that the authors should clarify:

1. The trend that is removed with their pivot detrending is linear, while there are indications that the real-world trend may be nonlinear at least in some parts of the world (e.g., Xu et al. 2022). The authors should discuss this issue and the possible implications of the linearity assumption.

We thank the reviewer for raising this important point.

We elected to perform the linear detrending because the trends for the NWS sub-regional JJA means are significant to the first (linear) term. We present the trends over the temporal period in a new figure below and now also include this figure as part of the Supplementary Information. Given the scatter inherent in the data, we opted for the simplest model that can explain the relationship well ($p < 0.01$). We have added additional text to the manuscript to further clarify this point and make reference to the new supplementary figure:

Lines 92-98: "This is because there are significant **linear** historical warming trends in the Celtic Sea, central North Sea and Irish shelf sub-regions (≈ 0.022 °C, ≈ 0.040 °C and ≈ 0.021 °C per year, respectively; **Figure S1**) which must be accounted for to make the distributions reflective of present-day climate."

Figure S1 – European North-West shelf seas warming rates. OSTIA JJA SST (black scatter markers) across the period 1982-2024 in the Celtic Sea (a), central North Sea (b) and Irish Shelf (c) sub-regions. The OSTIA linear regression fit is shown by the black line, and the associated slope, r and p values for the fit are annotated by text in each panel. Dark grey shading highlights the 1993-2016 hindcast period. Red markers mark the final point on the OSTIA linear fit (used for “last differencing”; see Methods).

2. While the UNSEEN approach is used to assess present-day probabilities, this approach is based on simulations over the period 1993-2016. How can we be sure that the statistics of that period are adequate for present-day conditions?

We thank the reviewer for raising this important point on the statistics of different climate periods. We find no statistically significant difference between the mean, standard deviation, kurtosis or skewness between the period we use and the latest 2016-2024 period. We also believe the question of the statistics (e.g. variability) of different periods to be of second order importance to MHW evolution compared to the primary influence of warming trends (e.g. Oliver 2019, Xu *et al* 2022) in the presence of

constant levels of climate variability, which we focus on in detail in our study (the OSTIA trend is specifically selected here for reference-year detrending because it is available until the present day). Finally, the short time between the 1933-2016 period and the present day provides us with confidence that the statistics of both periods will be similar enough for the purposes of our investigation.

Nevertheless, we acknowledge this is an important consideration and therefore add the following text to the manuscript to address the point more explicitly:

Lines 340-346: “The linear trend is removed in the reference-year detrended version but, importantly, the internal variability across the hindcast period is preserved. **For the purposes of our study, the variability in the hindcast period (1993-2016) is assumed to be an adequate analogue for the present day climate, due to i) the close temporal proximity between the periods and lack of significant differences in variability and because ii) the contribution of mean warming to global MHW trends is generally found to outweigh that of any changes to variability (Oliver 2019, Xu et al 2022).**”

3. The UNSEEN approach allows an evaluation of the probabilities for present-day conditions, effectively detrending the data. However, the authors also show that the likelihood of these extreme conditions has increased in recent years. If the effect of the mean state is removed from the present-day estimate, what causes the change in probabilities? Has internal variability changed? How? It would be important to include a discussion of this point.

The change in probabilities over the last 30 years is discussed in the manuscript as being because of a positive shift in the mean of a Gaussian distribution (rather than any change in width) of events (Lines 188-191). By iterating the year around which the data are reference-year detrended in Figures 3e and f (i.e. as a function of the warming trend), the procedure does indeed remove the trend across the hindcast period but also adjusts the distribution to be reflective of the climate of each year in turn, which effectively positively shifts the mean of each Gaussian across time. As the mean shifts positively, a greater number of events are shifted towards the extreme positive tail, hence a nonlinear increase in event statistics in response to a linear increase in the baseline. This effect is succinctly demonstrated in Figure 1 of Perkins (2015; copied below), itself adapted from Figure SPM.3 from IPCC (2012), which is also referenced on Line 191.

[figure redacted]

Figure 1 from Perkins (2015), itself adapted from IPCC (2012).

Itemized comments (mainly minor):

Caption of Fig. 1 and line 126. Here there is a mention to the 1993-2016 climatology. Before reading the Methods section it is not obvious why this period is chosen to define the climatology. It is only in the Methods section that we find out that this is the period covered by the initialized climate simulation, so I suggest that the use of this period is clarified either in the caption of Fig. 1 or in line 126 to avoid confusion.

Accepted and changed. Line 138 (previously Line 126) now reads: "... expressed as de-seasonalised anomalies against 1993-2016 climatology (**the model hindcast period**)."

Line 57-58 "upward trending frequency and intensity...". The upward trending frequency and intensity may depend on the definition of MHWs. If the definition is based on a fixed baseline the presence of a warming trend will lead to a more frequent and larger excursion of the SST anomalies above the fixed threshold as discussed in recent papers (e.g., Capotondi et al. 2024; Smith et al. 2025). Since the baseline choice seems central to the UNSEEN methodology, I suggest that the authors clarify this point here, and briefly discuss the issue of defining MHWs in a changing climate.

An advantage of using the UNSEEN methodology in this study is that we do not rely on a fixed threshold definition of a MHW but instead focus on events of similar magnitude of anomaly and duration as June-2023 ("June 2023-like") in large-sample distributions. However, the reviewer's point is important and discussed in the community. Therefore, we add text to the relevant sentence in the introduction to emphasise that the "upward trending frequency and intensity" mentioned is in the specific context of a fixed baseline.

Lines 63-64 now read: "... upward trending frequency and intensity of MHWs on the NWS (**with respect to a fixed baseline**; Cornes *et al* 2023, Simon *et al* 2023, Giménez *et al* 2024, Chen and Staneva 2024) ..."

Line 95. Should "stratifying" be "stratified"?

The choice of "stratifying" phrasing is intentional to be clear that the summer stratification is an annual phenomenon.

Line 166. Another useful reference here is Xu *et al.* (2022). This paper shows the key role played by the warming trend in the increasing frequency of MHWs. It also shows the influence of the nonlinearity of the trend on the assessment of changes in variability.

Thank you for highlighting this relevant reference and its applicability in this sentence. The reference is now included.

Line 168. The SST trend is not necessarily linear. Please discuss the implications of this assumption.

We refer to our response to major comment #1 above, which comments on the same matter.

Lines 187-189. This sentence is not clear to me. Maybe the content could be expanded and explained in different words or using an example.

We acknowledge that this sentence is unclear. We have decided to remove it and expand a previous sentence with related content to make our point more clearly.

Lines 212-218 now read: "UNSEEN fidelity tests suggest that distributions of the modelled JJA-14MAX timeseries means in the central North Sea (Celtic Sea) are somewhat cold (warm) biased relative to the observed statistic, **even if they are shown to be statistically indistinguishable according to the UNSEEN fidelity protocol**. In effect, the model may tend towards simulating lower strength events in the central North Sea than we may expect in reality. **This highlights an important point that fidelity tests classify the modelled distributions as being statistically indistinguishable from the observed climatology but cannot confirm that there are no biases whatsoever, due to the large and irreducible uncertainty in the observed climatology.** This could explain some of the difference between Celtic Sea and central North Sea probabilities extracted from the UNSEEN distribution. However, it is unlikely to be the full explanation."

Reviewer #2

The study by Atkins *et al.* aims to quantify the probability of the occurrence of marine heatwaves (MHWs) in summer on the European North-West shelf (NWS) seas with intensities similar to an unprecedented MHW event observed on the NWS in June 2023. To this end, the authors analyze model simulation output from a large initial condition ensemble of hindcast simulations by the UK Met Office GloSea model. An analysis protocol previously used for atmospheric extremes (UNSEEN) is applied to evaluate model fitness and to estimate probabilities of extreme events at specific event thresholds and warming levels. The authors find that MHWs similar in intensity to the event of June 2023 have a probability of around 10% to occur in any given year under current climate conditions, and that the increase in this probability is accelerating.

Overall, this paper presents novel and interesting findings about the probability of MHW events on the NWS. The application of the UNSEEN methodology to MHWs is a novel and valuable contribution to the field, and the findings should be of interest to regional stakeholders potentially affected by future extreme events. The methods and statistical analysis are well explained and reproducible. The text is concise, well written and clearly understandable, and the figures are well designed. Aside from the

minor comments detailed below, I believe this submission is suitable for publication in Communications Earth & Environment.

Thank you very much to the reviewer for the encouraging and constructive comments. A point-by-point response follows below:

Title: Since only one specific recent MHW event is analyzed, perhaps the title should not use the plural “heatwaves” (even though the results of course generalize to other MHWs of similar magnitude).

The choice of plural “heatwaves” is indeed intentional for the purposes of generalising to other MHWs in the current climate, as the reviewer states. In this sense, the results of our study are not *just* about the June 2023 event. For this reason, with respect, we prefer to keep the title as it currently is to maintain the link to the overall messaging of the paper.

Ln. 42: The introduction could put the 2023 NWS event into the context of the global record high temperatures observed at the same time (e.g., Huang et al. 2024; Terhaar et al. 2025; Cheng et al. 2024).

Thank you for this very useful suggestion. We have now added further context to the introduction. Lines 34-37 now read:

“Set in the context of a year of exceptional global climate anomalies in 2023 (Huang et al 2024, Terhaar et al 2025, Cheng et al 2024), the European North-West shelf seas (NWS) experienced an episode of notably high ocean temperatures, otherwise known as a marine heatwave (hereafter MHW), in June 2023...”

Ln. 44: Since this is the first line in the first figure’s caption, perhaps the abbreviations (NWS and/or MHW) could be spelled out.

Accepted and changed. The first line of the caption now reads: **“Figure 1 – The June 2023 European North-West shelf seas (NWS) marine heatwave...”**

Lns. 63–64: The term satellite era should be defined when it is first used, i.e. on line 37.

Thank you. Lines 39-40 now read: **“... which is unprecedented for the satellite era (approximately 1980s onwards; Berthou et al 2024)”**.

Lns. 84–85: The sentence after “Note, ...” is not very clear to me (although I acknowledge that there is a reference to the Methods), perhaps the detrending could be explained in simpler terms.

Thank you for raising this. We have adjusted the sentence to enhance clarity and keep the reference to the Methods where the process is explained in greater detail. Lines 89-92 now read: **“Note, the underlying SST data are detrended by fitting a linear least-squares trend and adjusting each point by the difference between its fitted value and the fitted value at the series end (2024). We refer to this procedure as “reference-year detrending” (see Methods)”**.

Ln. 105: Typo: “0For”

Thank you for pointing out this error. This has now been fixed.

Ln. 107: Missing word: “95% of the...”

Thank you for pointing out this error. This has now been fixed.

Ln. 107: It may be preferable to spell out the word “times” instead of using the symbol \times .

We assume that this comment refers to the use of the \times symbol on Line 161 in the original manuscript submission, rather than Line 107 where there is no use of the symbol. We have made the suggested change on Line 161.

Ln. 210: As in the introduction, the results could be put into the context of the global record air and sea surface temperatures observed in 2023.

The discussion has now been updated to draw comparisons in findings to an assessment of the 2023 global record (please see our response to Reviewer #3 concerning adding discussion of Terhaar et al., 2025). We also wish to politely flag the discussion of wider 2023 North Atlantic attribution literature already included in the manuscript (Lines 191-195), which we feel also addresses a related point even if about the North Atlantic region rather than global.

Lns. 243–244: Are there any existing studies using these hindcasts that could be cited?

Yes, there are relevant studies, and we have now added a sentence citing these papers. Lines 282-283 read: “**GloSea hindcast data have been used in previous UNSEEN studies in the context of atmospheric extremes (e.g. Wang et al 2020, Kent et al 2022)**”.

Ln. 255: Unclear what is meant by “climate-quality observations”, should this say e.g. “quality-controlled climate observations”?

We have now removed the “climate-quality” phrasing altogether given it is not strictly necessary and is unclear. Thank you for pointing this out.

Ln. 257: “this chapter” → this study?

Thank you for pointing out this error. This has now been fixed.

Ln. 281: It could be helpful to include a Supplementary Figure schematically showing the pivot detrending procedure.

In response to Reviewer #1’s major comment, we have added a new supplementary figure (Figure S1) which shows the trends across the observational period for each sub-region (copied below as well). We have now also added elements to this new figure which, alongside additions to the manuscript text (described next), help enhance clarity of the reference-year detrending procedure description.

Lines 331-334: “First, for each individual sub-region, we apply a linear regression fit across the 1982-2024 OSTIA JJA SST record (**black lines in Figure S1**). Next, the difference between each point on the trend line across 1993–2016 (matching the GloSea hindcast period) and the final point of the linear fit (2024, present day at time of writing; **red markers in Figure S1**) is calculated (‘last differencing’) ...”

Figure S1 – NWS warming rates. OSTIA JJA SST (black scatter markers) across the period 1982-2024 in the Celtic Sea (a), central North Sea (b) and Irish Shelf (c) sub-regions. The OSTIA linear regression fit is shown by the black line, and the associated slope, r and p values for the fit are annotated by text in each panel. Dark grey shading highlights the 1993-2016 hindcast period. Red markers mark the final point on the OSTIA linear fit (used for ‘last differencing’; see Methods).

Ln. 299: It would be interesting to include a version of Fig. 2 using the alternative SST trend from GloSea as a Supplementary Figure.

We agree this is an interesting point to acknowledge via a new supplementary figure. However, rather than repeating Figure 2 using the alternative SST trend from GloSea (which produces effectively the same figure with the observed statistic lying within the central 95% of the distribution), we believe it is more informative to add a figure which demonstrates that the observed trend is statistically indistinguishable from the model ensemble trends. In other words, that the observed trend can be reliably drawn from the ensemble trend distribution.

The new supplementary figure is copied below:

Figure S2 – Modelled vs. observed NWS SST warming trends. Frequency distributions of proxy model SST trends ($n = 10,000$) in the Celtic Sea (a), central North Sea (b) and Irish Shelf (c) sub-regions. The observed SST trend is marked by black dashed lines. The percentile value at which the observed statistic lies within the modelled distribution is marked by text annotation. The proxy model SST timeseries ($n = 10,000$), from which the trends are calculated and the modelled distributions built, are generated by sampling the full model ensemble by bootstrapping (with replacement) across the hindcast period. The timeseries span the common period between model and observations (i.e. 1993-2016 hindcast period).

Furthermore, we add the following text to the manuscript which makes reference to the new supplementary figure:

Lines 354-358: “In any case, the OSTIA and GloSea-derived trends are **statistically indistinguishable (observed trend lies within the central 95% of the model trend distribution; Figure S2)** and result in discrepancies of just ± 1 -2 percentage points in the final June 2023-like probability estimates for the present day when used for the reference-year detrending (Figure 3 versus Figure S3)”.

Finally, please see our response to Reviewer #3 minor comment re: Line 307 for more detail on the effect of the choice of trend on JJA-14MAX probabilities.

General: The manuscript does not yet contain a data and code availability statement.

Thank you for bringing this to our attention. We have now included Data and Code availability statements at the end of the manuscript, in line with the requirements of Nature Portfolio journals.

Reviewer #3

This work explores how the likelihood of marine heatwaves (MHWs) in the European North-Western Shelf (NWS) has changed with time and quantifies the probability of a particular record-breaking event occurring. The foundation of this work is the “UNSEEN” method, which builds a distribution of potential MHW events from physically plausible climate model simulations. The UNSEEN methodology is validated, here and elsewhere. The main results show how an event of the magnitude of June 2023 has become more likely, and continues to become more likely, with time over the past 3 decades.

The research is engaging and the content well written. I enjoyed the succinct nature of the paper, although I find that also means some explanations and justifications are lacking. In particular, the UNSEEN methodology needs to be explained carefully. The main conclusion, on the increased likelihood of intense MHWs, is somewhat obvious, but I appreciate that the quantification is an important step. The authors communicate fairly well the need for such studies, but I feel not yet fully

convinced that the quantification alone justifies publication in this journal. Before publication some clarifications need to be made, in my opinion.

Thank you for your detailed and constructive comments. We provide a point-by-point response below:

Main Comments

In the abstract, some statements are unclear. “The probability of June 2023-like events occurring in the present day is approximately 10% in any given year”; given the definition of a MHW as an exceedance of the 90th percentile, in a detrended dataset, the probability of an event is always 10%. Also, what do you mean by “any given year”? I thought the chance increased with time (thanks to the trend).

The exceedance of the 90th percentile is indeed a commonly used definition of a MHW in the literature. However, in this study our reference is not a percentile threshold defined across a climate period but rather we mine for events in the UNSEEN distribution which are of similar magnitude of anomaly and duration as June-2023 (“June 2023-like”).

Regarding the point on the “any given year” phrasing, we have rearranged the relevant sentence in the abstract to enhance clarity. Lines 25-27 now read: “Here, by employing a large ensemble of initialised climate model simulations, we show that the probability of June 2023-like events occurring is approximately 10% **in any given year of the present-day climate**”.

If this was an unprecedented, what was the next closest event i.e. what is the next largest JJA-14MAX on record?

By our calculations, the next largest NWS-wide mean JJA-14MAX event across the OSTIA 1982-2024 record is +1.74 °C in 2009 (relative to 1993-2016 climatology), compared to +2.13 °C in 2023. We have added text to further contextualise the 2023 event and demonstrate the extent to which it exceeded the previous record:

Lines 137-142: **“Real-time observational estimates recorded the NWS-wide peak 14-day rolling mean SST anomaly in June 2023 at +2.13 °C, expressed as de-seasonalised anomalies against 1993-2016 climatology (the model hindcast period). This was unprecedented in the satellite era and substantially higher than the previous record (+1.74 °C in 2009). At the sub-region level, 2023 JJA-14MAX anomalies were recorded at +3.12 °C and +2.56 °C for the central North Sea and Celtic Sea, respectfully”**.

Recent and similar high-profile work has been published on “unlikely but not unexpected” SST jumps. How does this compare to UNSEEN? Terhaar, J., Burger, F.A., Vogt, L. et al. Record sea surface temperature jump in 2023–2024 unlikely but not unexpected. *Nature* 639, 942–946 (2025). <https://doi.org/10.1038/s41586-025-08674-z>

We thank the reviewer for raising this important recent literature (similarly to Reviewer #2). We have made the following additions to the discussion section where we place our results in the context other recent attribution literature.

Lines 191-209: “In further support of our findings, recent work by Guinaldo *et al* (2025) applies a different attribution method, namely counterfactual simulations using unconstrained model ensembles, but for the wider North Atlantic basin, and similarly argues that 2023-type events should be expected today (\approx 10-year return period) in the context of anthropogenic warming. Berthou *et al*. (2024) also demonstrate a contribution of the warming trend to the intensity of SST anomalies during the NWS June 2023 MHW and that such levels are projected to become commonplace past the mid-century under RCP8.5 conditions. **On a global scale, Terhaar *et al* (2025) show using a large sample of observation-based synthetic SST timeseries that record-breaking anomalies in 2023-2024 would**

have been practically impossible without a global warming trend. Our study complements and builds upon previous **attribution** studies by using observationally constrained models to provide actionable near-term information which will be critical to immediate on-shelf marine planning and management. Moreover, to our best knowledge, the accelerating rate of change in probability of unprecedented NWS MHWs over recent decades is shown here for the first time.”

Trends are an important part of the discussion – please consider adding maps in the supplementary.

We believe that the new supplementary figure added in response to Reviewer #1 and Reviewer #2’s comments (i.e. Figure S1) addresses this suggestion as well. Note, however, that these are included as scatter/line plots per sub-region, rather than maps, which we find to be more useful for visualising the linear nature of the trend for each sub-region (see response to Reviewer #1 major comment #1).

The authors claim to use a dataset of “plausible...events which could occur in the present day”. The benefit of this dataset is that it uses a very large ensemble. However, it is based on one model. That the representation of events changes between models should be considered. The authors should explain why using a larger ensemble from one model is better than using a potentially equally-large ensemble from a range of models (i.e. the other Copernicus seasonal forecasting systems).

We agree with the principle that using multi-model ensembles are worthwhile for this kind of analysis. However, we feel it is important to note here that it is difficult to ascertain which models from the ensemble perform best in relation to observations for attribution purposes because the observational record is limited and itself contains large irreducible uncertainty. In this sense, ensuring a large number of samples, even if from the same model, is in our view of first-order importance and sourcing realisations from different models is second-order.

Once again, though, we do agree that using multi-model approaches is a valid and worthwhile future direction. Whilst it is beyond the scope of our study, we have added text to the discussion to directly acknowledge this point.

Lines 282-285 : “GloSea hindcast data have been used in previous UNSEEN studies in the context of atmospheric extremes (e.g. Wang *et al* 2020, Kent *et al* 2022). **Using a multi-model ensemble for UNSEEN analysis (e.g. Jain *et al* 2020) would reduce sensitivity to model structure (Kelder *et al* 2022) and represents a worthwhile step for future work, but is beyond the scope of the present study”.**

Where ensemble members agree (in regions of high memory/persistence) the UNSEEN model is not used. The authors state that the effective sample size decreases. Is it not that the distribution simply narrows? Surely the number of samples is the same?

Yes, indeed the number of samples is still the same. This is what is meant when we state how the *effective* sample size is reduced (using the same terminology as Kelder *et al* 2022).

Regarding the fidelity tests, falling with the central 90% of the distribution seems a very easy target. I notice that if the threshold were 90%, then the Celtic Sea would be removed too. Please justify this better.

We select the central 95% criterion because of the precedent set in previous literature. We have rearranged the relevant passage to make this justification clearer:

Lines 310-315 : “... We sample the full GloSea ensemble across the hindcast period by bootstrapping (with replacement) to produce 10,000 proxy timeseries of the same length as the observed record over 1993–2016 (common period between model and observations). When the single observed (OSTIA) statistic for each statistical moment lies within the central 95% of the modelled distribution (i.e. within

the 2.5th and 97.5th percentiles), we consider the model to be statistically indistinguishable from the observations and therefore suitable for UNSEEN analysis. **This approach mirrors that taken in previous studies which apply the UNSEEN approach to climate extremes (Thompson *et al* 2017, 2019, Kay *et al* 2020, 2023, Kelder *et al* 2020, Cotterill *et al* 2024)**".

Authors effectively communicate the need for risk assessments and such quantifications. How does the UNSEEN method compare to attribution studies which also aim to provide how likelihoods of events change with climate change?

In response to this comment, we point to the discussion already included comparing our results to other relevant attribution studies (Lines 190-195) as well as specifically the new inclusion of the reference and discussion of the recent Terhaar *et al.* (2025) study in response to the reviewer's previous comment.

This study focuses on a single NWS MHW, so the inclusion of "Recent... heatwaves" in the title is inaccurate.

Please see our response to Reviewer #2's first comment.

Minor Comments

Line 153: Why is the growth non-linear? Is the background SST trend non-linear?

The background SST trend is linear (please see our response to Reviewer #1's first major comment for more detail here and action taken). The growth of event likelihood is nonlinear due to the effect of a positive mean shift in a Gaussian distribution of events (please see our response to Reviewer #1's third major comment for more detail).

Line 230: Is GloSea run in burst or lagged mode? Or is this a non-operational version?

GloSea is run in lagged mode. Thank you for raising this point which can be clarified. We have added the following detail to the manuscript:

Lines 273-277: "Seven individual members [...] are initialised across three start dates centred on the 1st of the month preceding the summer (JJA) season (25th April, 1st May, 9th May; **i.e. lagged mode**) and are combined to obtain an ensemble of simulations at 2-4 months lead time for each hindcast year.

Line 231: Typically new releases have a range of differences. Is the only difference between 6 and 5 the model? No change in resolution, initial conditions?

Yes, the only primary difference between GloSea5 and GloSea6 at this point is the underlying model. The GloSea6 release is taking place in multiple phases. A future phase will indeed increase the ensemble size but this is not used here.

Line 244: What are these multiple hindcasts? How do they differ?

The hindcasts simply differ by when (in real life) they were run operationally at the UK Met Office. GloSea has been run over the same hindcast period multiple times in different (real life) years. Due to stochastic perturbations to initial conditions all members across different hindcast runs are independent. In other words, they can act as additional ensemble members for the same hindcast period.

Line 246: Where does the factor of two (from the two versions, 5 & 6) come into the calculation of 2520?

Of the five hindcast ensembles employed, two are from GloSea5 and three are from GloSea6. Therefore, together they constitute the “5 hindcasts” in the calculation: 5 hindcasts × 3 start dates × 7 members × 24 years = 2520 samples.

Line 290: The last point being 2016?

The last point of the linear fit is 2024. The important detail here is that we are discussing using points on the linear trend line vs. using the raw data point which is actually recorded from 2024, rather than which year is being discussed. We have edited the sentence to avoid confusion:

Lines 335-336: “We use the **2024 point** on the linear fit, as opposed to the actual 2024 data point, to produce a less noisy estimate for 2024.”

Line 307: I do not think “not shown” is allowed in this journal. Anyway, I recommended including trend maps.

Thank you for pointing this out, we have now removed “not shown” and replaced it with a reference to a final supplementary figure (copied below), which shows an equivalent analysis to Figure 3 but using the GloSea ensemble trend for the reference-year detrending.

Lines 354-358: “In any case, the OSTIA and GloSea-derived trends are statistically indistinguishable (observed trend lies within the central 95% of the model trend distribution; Figure S2) and result in discrepancies of **approximately** ± 1 -2 percentage points in the final June 2023-like probability estimates for the present day when used for the reference-year detrending (**Figure 3 versus Figure S3**)”.

Figure S3 – UNSEEN MHWs exceeding the June 2023 event (reference-year detrended using model ensemble trend). For the Celtic Sea (red; panel column a, c, e) and central North Sea (blue; panel column b, d, f): UNSEEN distribution of modelled JJA-14MAX events (a, b), UNSEEN estimate of event chance as a function of degrees Celsius above June 2023 (c, d) and UNSEEN estimate of June 2023 event chance as a function of time since 1993 (e, f). Observed peak 14-day rolling mean SST in June 2023 is marked by the black dashed lines in a) and b), and the probability of this event in the UNSEEN distribution (i.e. 100% minus percentile rank) is shown by text annotation. In c) – f), probabilities are also calculated relative to observational estimates, shading represents the 95% confidence interval calculated by bootstrapping ($n = 1000$). The UNSEEN reference-year detrending approach for 2024 climate (here, calculating the trend from the model ensemble mean, extrapolated to 2024; see Methods **Reference source not found.**) is applied in all panels except e) and f) where probabilities are calculated from distributions which are reference-year detrended to reflect the climate of each year in turn. The vertical black dashed lines in e) and f) mark the year 2023, i.e. when the June 2023 event actually occurred.

References

- Cheng L, Abraham J, Trenberth K E, Boyer T, Mann M E, Zhu J, Wang F, Yu F, Locarnini R, Fasullo J, Zheng F, Li Y, Zhang B, Wan L, Chen X, Wang D, Feng L, Song X, Liu Y, Reseghetti F, Simoncelli S, Gouretski V, Chen G, Mishonov A, Reagan J, Von Schuckmann K, Pan Y, Tan Z, Zhu Y, Wei W, Li G, Ren Q, Cao L and Lu Y 2024 New Record Ocean Temperatures and Related Climate Indicators in 2023 *Adv. Atmospheric Sci.* **41** 1068–82
- Huang B, Yin X, Carton J A, Chen L, Graham G, Hogan P, Smith T and Zhang H-M 2024 Record High Sea Surface Temperatures in 2023 *Geophys. Res. Lett.* **51** e2024GL108369
- IPCC 2012 Managing the Risks of Extreme Events and Disasters to Advance Climate Change Adaptation *Special Report of the Intergovernmental Panel on Climate Change* ed C B Field, V Barros, T F Stocker and Q Dahe (Cambridge University Press) p 593
- Jain S, Scaife A A, Dunstone N, Smith D and Mishra S K 2020 Current chance of unprecedented monsoon rainfall over India using dynamical ensemble simulations *Environ. Res. Lett.* **15** 094095
- Kent C, Dunstone N, Tucker S, Scaife A A, Brown S, Kendon E J, Smith D, McLean L and Greenwood S 2022 Estimating unprecedented extremes in UK summer daily rainfall *Environ. Res. Lett.* **17** 014041
- Oliver E C J 2019 Mean warming not variability drives marine heatwave trends *Clim. Dyn.* **53** 1653–9
- Perkins S E 2015 A review on the scientific understanding of heatwaves—Their measurement, driving mechanisms, and changes at the global scale *Atmospheric Res.* **164–165** 242–67
- Terhaar J, Burger F A, Vogt L, Frölicher T L and Stocker T F 2025 Record sea surface temperature jump in 2023–2024 unlikely but not unexpected *Nature* **639** 942–6
- Wang L, Hardiman S C, Bett P E, Comer R E, Kent C and Scaife A A 2020 What chance of a sudden stratospheric warming in the southern hemisphere? *Environ. Res. Lett.* **15** 104038
- Xu T, Newman M, Capotondi A, Stevenson S, Di Lorenzo E and Alexander M A 2022 An increase in marine heatwaves without significant changes in surface ocean temperature variability *Nat. Commun.* **13** 7396

Review of COMMSENV-25-2206-T

Communications Earth & Environment

The study by Atkins et al. aims to quantify the probability of the occurrence of marine heatwaves (MHWs) in summer on the European North-West shelf (NWS) seas with intensities similar to an unprecedented MHW event observed on the NWS in June 2023. To this end, the authors analyze model simulation output from a large initial condition ensemble of hindcast simulations by the UK Met Office GloSea model. An analysis protocol previously used for atmospheric extremes (UNSEEN) is applied to evaluate model fitness and to estimate probabilities of extreme events at specific event thresholds and warming levels. The authors find that MHWs similar in intensity to the event of June 2023 have a probability of around 10% to occur in any given year under current climate conditions, and that the increase in this probability is accelerating.

Overall, this paper presents novel and interesting findings about the probability of MHW events on the NWS. The application of the UNSEEN methodology to MHWs is a novel and valuable contribution to the field, and the findings should be of interest to regional stakeholders potentially affected by future extreme events. The methods and statistical analysis are well explained and reproducible. The text is concise, well written and clearly understandable, and the figures are well designed. Aside from the minor comments detailed below, I believe this submission is suitable for publication in Communications Earth & Environment.

Title Since only one specific recent MHW event is analyzed, perhaps the title should not use the plural “heatwaves” (even though the results of course generalize to other MHWs of similar magnitude).

Ln. 42 The introduction could put the 2023 NWS event into the context of the global record high temperatures observed at the same time (e.g., Huang et al. 2024; Terhaar et al. 2025; Cheng et al. 2024).

Ln. 44 Since this is the first line in the first figure’s caption, perhaps the abbreviations (NWS and/or MHW) could be spelled out.

Lns. 63–64 The term *satellite era* should be defined when it is first used, i.e. on line 37.

Lns. 84–85 The sentence after “Note, ...” is not very clear to me (although I acknowledge that there is a reference to the Methods), perhaps the detrending could be explained in simpler terms.

Ln. 105 Typo: “0For”

Ln. 107 Missing word: “95% of the...”

Ln. 107 It may be preferable to spell out the word “times” instead of using the symbol \times .

Ln. 210 As in the introduction, the results could be put into the context of the global record air and sea surface temperatures observed in 2023.

Lns. 243–244 Are there any existing studies using these hindcasts that could be cited?

Ln. 255 Unclear what is meant by “climate-quality observations”, should this say e.g. “quality-controlled climate observations”?

Ln. 257 “this chapter” → this study?

Ln. 281 It could be helpful to include a Supplementary Figure schematically showing the pivot detrending procedure.

Ln. 299 It would be interesting to include a version of Fig. 2 using the alternative SST trend from GloSea as a Supplementary Figure.

General The manuscript does not yet contain a data and code availability statement.

References

- Huang, Boyin et al. (2024). “Record High Sea Surface Temperatures in 2023”. In: *Geophysical Research Letters* 51.14, e2024GL108369. ISSN: 1944-8007. DOI: 10.1029/2024GL108369.
- Terhaar, Jens et al. (2025). “Record Sea Surface Temperature Jump in 2023–2024 Unlikely but Not Unexpected”. In: *Nature* 639.8056, pp. 942–946. ISSN: 1476-4687. DOI: 10.1038/s41586-025-08674-z.
- Cheng, Lijing et al. (2024). “New Record Ocean Temperatures and Related Climate Indicators in 2023”. In: *Advances in Atmospheric Sciences* 41.6, pp. 1068–1082. ISSN: 1861-9533. DOI: 10.1007/s00376-024-3378-5.